# Enhancing authenticity, diagnosticity and *e*quivalence (AD-Equiv) in multicentre OSCE exams in health professionals education: protocol for a complex intervention study

Peter Yeates,[1] Adriano Maluf [iD],[1] Ruth Kinston,[1] Natalie Cope,[1] Gareth McCray,[1] Kathy Cullen,[2] Vikki O'Neill,[2] Aidan Cole,[2] Rhian Goodfellow,[3] Rebecca Vallender,[3] Ching-Wa Chung,[4] Robert K McKinley [iD],[1] Richard Fuller,[5] Geoff Wong [iD] [6]

For numbered affiliations see end of article.

**Correspondence to**
Dr Peter Yeates;
p.yeates@keele.ac.uk

## ABSTRACT

**Introduction** Objective structured clinical exams (OSCEs) are a cornerstone of assessing the competence of trainee healthcare professionals, but have been criticised for (1) lacking authenticity, (2) variability in examiners' judgements which can challenge assessment equivalence and (3) for limited diagnosticity of trainees' focal strengths and weaknesses. In response, this study aims to investigate whether (1) sharing integrated-task OSCE stations across institutions can increase perceived authenticity, while (2) enhancing assessment equivalence by enabling comparison of the standard of examiners' judgements between institutions using a novel methodology (video-based score comparison and adjustment (VESCA)) and (3) exploring the potential to develop more diagnostic signals from data on students' performances.

**Methods and analysis** The study will use a complex intervention design, developing, implementing and sharing an integrated-task (research) OSCE across four UK medical schools. It will use VESCA to compare examiner scoring differences between groups of examiners and different sites, while studying how, why and for whom the shared OSCE and VESCA operate across participating schools. Quantitative analysis will use Many Facet Rasch Modelling to compare the influence of different examiners groups and sites on students' scores, while the operation of the two interventions (shared integrated task OSCEs; VESCA) will be studied through the theory-driven method of Realist evaluation. Further exploratory analyses will examine diagnostic performance signals within data.

**Ethics and dissemination** The study will be extra to usual course requirements and all participation will be voluntary. We will uphold principles of informed consent, the right to withdraw, confidentiality with pseudonymity and strict data security. The study has received ethical approval from Keele University Research Ethics Committee. Findings will be academically published and will contribute to good practice guidance on (1) the use of VESCA and (2) sharing and use of integrated-task OSCE stations.

## STRENGTHS AND LIMITATIONS OF THIS STUDY

⇒ The study uses a complex intervention design to explain how two separate interventions operate when jointly shared across medical schools to address authenticity and equivalence: (1) integrated-task objective structured clinical exam (OSCE) stations and (2) video-based examiner score comparison and adjustment.

⇒ The study's multicentre design provides broadly sampled insight into the operation of integrated-task OSCE stations across different contexts.

⇒ Use of Realist Evaluation will give rich insight into how these interventions work or do not work, under what circumstances, for whom and why.

⇒ Video-based comparison of examiners' scoring will provide controlled comparisons between schools of a subset of examiners' scoring, thereby enabling appraisal of the likelihood of bias arising from inter-institutional differences in implementation.

## INTRODUCTION

Dependable assessment of the performance and skills of graduating health professionals (doctors, nurses, physiotherapists, pharmacists, etc) remains critical to ensuring fairness for students[1] and patient safety.[2 3] Objective structured clinical exams (OSCEs) generally involve students rotating around a carousel of timed, simulated clinical tasks being observed on each task by different, trained, examiners who score performances using specified criteria.[4] Over recent decades, OSCEs have become one of the preeminent methods of assessing clinical skills performance[5] due to their ability to ensure students are directly observed[6] under equivalent conditions[7] according to an appropriate assessment blueprint[8] while avoiding some of the limitations of workplace assessments such as case

BMJ

selection, impression management,[9] or prior performance information.[10]

Despite these benefits, OSCEs have been criticised for:
► Lacking authenticity.
► Examiner variability, which can challenge equivalence.
► Limited ability to ensure that students are competent in all skills domains.

The authenticity of OSCEs has been criticised due to their simulated context and task fragmentation,[11 12] which in turn could challenge the applicability of their outcomes to clinical practice.[13] In response, several institutions have explored use of OSCE stations which combine multiple tasks[14 15]—termed 'integrated task OSCEs' or greater levels of simulation fidelity[16] to more closely mimic real practice. While these appear to offer a promising development, it is unclear how the added complexity of these tasks influences examiners' judgements and therefore OSCE standardisation.

Furthermore, examiner variability in OSCEs continues to be significant.[17] Owing to student numbers, OSCE exams are often run across several ostensibly identical parallel versions of the same exam or distributed across geographical locations, with different examiners in each parallel version. Several studies suggest potentially important differences between the different cohorts of examiners in each parallel version of the exam within single institutions[18] or in large-scale distributed exams.[19 20] While these variations could compromise the fairness or safety of the resulting assessment decisions, they are rarely studied due to difficulties in directly measuring the influence of unlinked groups of examiners in different parallel versions of the exam. Consequently, little is known about how regional variations in examiners' judgements might challenge the equivalence of OSCEs[21] which could produce different outcomes for students in OSCE exams.

Two prerequisites are necessary to determine equivalence within a distributed OSCE: first, common (or shared) OSCE content is needed, in order for examiners' judgements to be comparable, and second, a method is needed to compare examiners' scoring when they are distributed across different locations. In the UK, medical schools set their own OSCE exams, resulting in variation in content and format between Schools. Consequently, sharing OSCE content between schools, while necessary, will involve change from usual practice which could further influence examiner variability or produce unintended consequences.

Recently, Yeates et al[22–24] have iteratively developed a method to compare examiners' scoring within distributed OSCEs, called video-based examiner score comparison and adjustment (VESCA). This produces linking of otherwise unlinked groups of examiners (termed 'examiner-cohorts'[18] by (1) videoing a small subset of students on each station of the OSCE; (2) asking examiners from all examiner cohorts to score the same station-specific comparator videos; and (3) using the resulting score linkage to compare and equate for differences in examiner-cohorts. Their findings suggest that despite

following accepted procedures for OSCE conduct, significant differences may persist between groups of examiners which could affect the pass/fail classification of a significant minority of students. Follow-up work has enhanced the technique's feasibility,[24] and shown that it is adequately robust to several potential confounding influences[25] and variations in implementation.[26] While these findings suggest that examiner-cohort effects are important and support the validity of VESCA for their measurement, VESCA has not yet been used across institutions, so both the likely magnitude of effects which may arise, and the practical implications of applying the method across institutions are unknown.

Finally, recent inquiry has focused on ensuring that trainees are competent across all relevant domains of performance,[27] with a view to both providing diagnostic information to support their learning and enabling focused areas of deficit to be addressed rather than simply demonstrating a sufficient total score, as is often the case in OSCEs.[28] This had led to scrutiny of the ability of OSCEs to prevent compensation between domains[29] and whether OSCEs could provide greater diagnosticity of students' areas of focal weakness. While non-compensatory domain-based scoring has been trialled in other arenas,[30] little is known about the psychometric properties of such domain scores or whether they can provide independently reliable scores for the constructs they represent. As the utility of VESCA would be greatly enhanced by providing domain level information which has been adjusted for the examiner-cohort effects, it is desirable to study the potential for these data to provide that information.

Collectively, it is anticipated that if these interventions are able to enhance the authenticity and equivalence of OSCEs while providing more diagnostic information on learners' performance, this will enhance OSCEs ability to support learning through their influence on students' preparation for OSCEs and their subsequent provision of more diagnostic feedback, while also ensuring greater confidence in the progression decisions which they inform. Consequently, understanding the interaction and use of these innovations is critical to determining their ability to benefit educational and healthcare practice.

## Aims and objectives

This project has a series of aims, objectives and research questions that set out to address the criticisms described above about OSCE examinations. These are:

### Criticism 1: lack of authenticity
► Objective 1: to increase perceived authenticity of an OSCE through use of integrated-task OSCE stations.

### Criticism 2: examiner variability and challenges to equivalence
► Objective 2: to share integrated-task OSCE stations across different institutions and understand the implications which arise from the interaction of these

stations with existing individual perceptions and institutional assessment practices.

Then, developing from that objective

► Objective 3: to use the VESCA methodology, within the context of a multicentre integrated-task OSCE, to

a. compare and equate for differences between examiner cohorts in different institutions and

b. understand the implications which arise from using VESCA across institutions.

### Criticism 3: limited diagnosticity of OSCEs across different domains of performance

► Objective 4: to determine whether different subdomains of performance can be reliably distinguished from each other (rather than only providing an overall competence score) within a shared integrated-task OSCE.

### Research questions
#### Objectives 1 and 2 will be addressed jointly through research question 1

When integrated-task stations are used and shared within an OSCE, how, when, why and to what extent do examiners, students and simulated patients use and interact with them and how does this influence their perception of the authenticity of the OSCE scenarios?

#### Objective 3a will be addressed by the following research questions 2–5

► How does the standard of examiners' judgements compare between examiner cohorts?

► How does the standard of examiners' judgements compare between institutions?

► What are the relative magnitudes of interinstitutional versus intrainstitutional variation?

► How much influence does adjusting for examiner-cohort effects have on students':
  a. Overall scores.
  b. Categorisation (fail/pass/excellence).
  c. Rank position.

#### Objective 3B will be addressed through research question 6

When VESCA is used to compare and equate for differences between examiner cohorts in different institutions within the context of a shared integrated-task OSCE, how, when, why and to what extent do examiners, students and simulated patients use and interact with VESCA?

#### Objective 4 will be addressed through research questions 7–8

► How reliably can different domains of assessment be discriminated in unadjusted data?

► Do students show differing patterns of performance across different domains of the assessment in unadjusted data?

### METHODS
#### Methodological overview

The study will use a complex intervention design[31] to implement VESCA in the context of a multicentre authentic-task OSCE. Research approaches will comprise psychometric analysis of assessment data[32] and Realist evaluation,[33] collecting data through mixed methods. A schematic overview of the data collection and analysis is provided in figure 1.

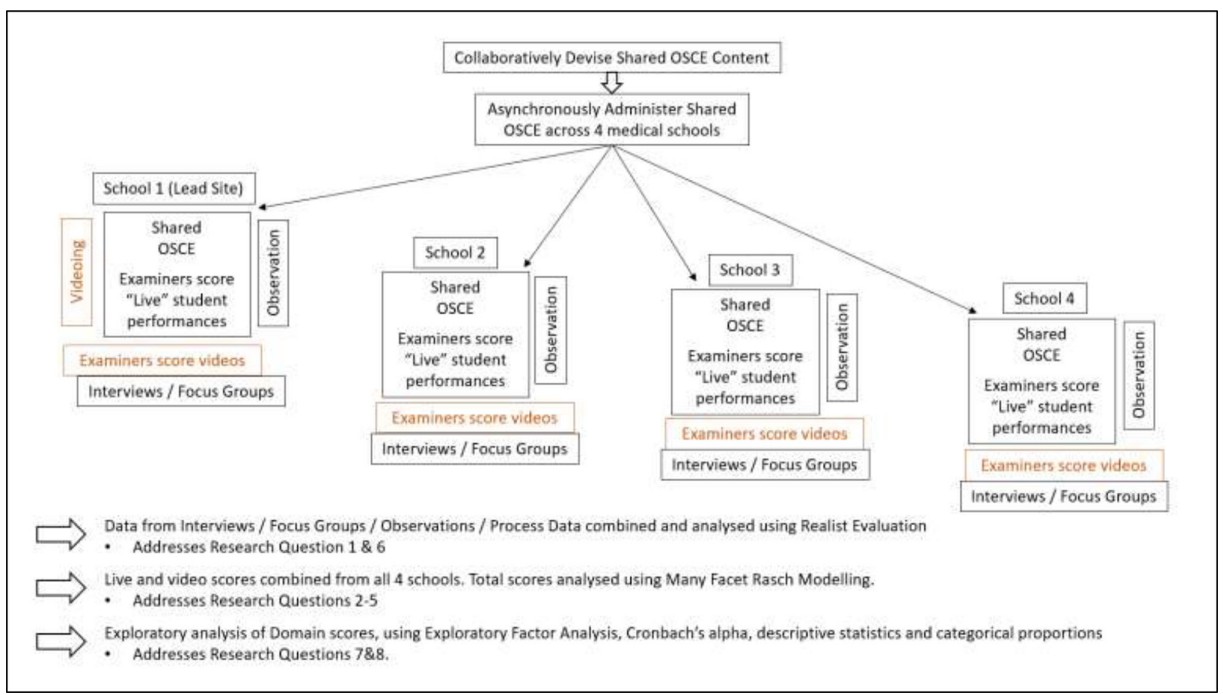

**Figure 1** Schematic of the data collection and analysis processes. OSCEs, objective structured clinical exams.

## Population, sampling and recruitment

The study population will comprise participants of late years (penultimate and final year) undergraduate medical student clinical exams within the UK.

This population will be sampled by recruiting four medical schools to participate as centres in the study, with sampling from all relevant examiners, students, simulated patients. As no prior work has formally compared OSCE examination standards across UK medical schools, the study will aim to sample across different characteristics which might plausibly influence scoring: geographic divergence; Russell group and non-Russell group Universities; and new and more established medical schools.

Recruitment will be performed locally by each participating institution using both in-person and electronic advertisements. Each participating institute will have recruitment targets for students (n=24), examiners (n=12) and simulated patients (n=12). This sample size is pragmatic based on the resource implications for individual institutions of running a research OSCE. While no formal method exists to power comparisons, or any agreed minimally important difference for differences between groups of OSCE examiners, subset analysis of data from[24 24] suggests that this sample size is likely to provide a SE in the region of 0.03 logits, enabling statistically significant detection of a difference between examiner cohorts of 5% of the assessment scale.

## OSCE design

The OSCE will comprise six tasks (stations). In each station, students will be directly observed for 13.5 min, with a further variable amount of preparation and rotation time of between 1.5 and 4 min per station, depending on each school's usual practice. Consequently, total testing time will range between 90 and 105 mins depending on different schools' practices.

Station content (simulated patient scenarios/instructions/stimulus materials/scoring rubrics) will be developed by the research team to reflect plausible simulated scenarios from foundation year 1 doctors routine work and integrate multiple related processes which would be required for whole-task completion. For example, a station may describe a specific clinical scenario from the work of a new doctor and instruct candidates to perform a relevant clinical assessment. Candidates might then be expected to gather a clinical history, perform relevant focused physical examination, interpret provided investigation results, consult available guidelines and then describe their diagnosis and management to the patient. Tasks will be blueprinted against the UK General Medical Council's Clinical Skills Performance Assessment framework,[34] to sample this framework's three domains: areas of clinical practice; clinical and professional capabilities; and areas of professional knowledge. The same stations will be used in all four study sites, while allowing minor adaptation for local contexts (for example by providing local antibiotic guidelines or dosage calculators).

Individual students will rotate around all six stations, and be observed by a different, single examiner in each station during a 90 min 'cycle' of the exam. Each site will host two parallel circuits of the OSCE (identical OSCE stations, run with different examiners). Twelve students will be examined in each parallel circuit (ie, 2 cycles of 6 students), enabling 24 students to be tested at each site.

Examiners will be provided with station material (clinical scenario, simulated patient script, marking criteria) prior to the OSCE. Additionally, examiners will be provided with a weblink to a training video which will orientate them to the scoring format.

Examiners at all sites will score students' performances on the GeCoS rating system.[35] This scoring system selects 5 appropriate performance domains for each station from a list of 20 when the station is designed (eg, history content, physical examination, clinical reasoning, building and maintaining the relationship, management content). Each domain is scored 1–4 (1=must improve; 2=borderline; 3=proficient; 4=very good). These scores are combined with a further 7-point global rating (1=incompetent; 7=excellent) to give a total score out of 27 for each station. Scoring will use tablet-based or paper-based marking based on available resources at each site.

The OSCE will be conducted first at the lead site (Keele) to enable video production for VESCA procedures; timing in other institutions will vary within an 8-month window to fit with local curricular demands. Local site teams will operationalise the station content based on the constraints of their local resources and equipment. Timing of stations will use local timing facilities but will adhere to standard timing intervals.

## Intervention

VESCA will be employed using the methods developed by Yeates et al.[22–24]

*Video filming:* performances of all students in all six stations, from the first cycle, on a selected circuit, will be filmed at the lead site (Keele) using methods established by Yeates et al.[23] Filming will use two unobtrusive wall-mounted closed-circuit TV cameras in every room (ReoLink 432, 1080 HD resolution). Camera position, angle and zoom will be selected to optimise capture of the performance. Sound will be recorded using a stereo condensing boundary microphones (Audio-Technica Pro 44). The first three videos from each station which are technically adequate (unobstructed pictures with adequate sound) will be selected and processed for further use, resulting in three comparison videos for each of the six stations in the OSCE.

Video scoring: all examiners will be asked to score the three selected videos selected for the station they examine. All examiners who examine a given station will score the same videos. To facilitate this, videos will be securely shared across institutions, using the secure online video scoring approach developed by Yeates et al.[24] This will include the following elements: online consent; station-specific examiner information; sequential presentation of

the three comparison videos for the station. Examiners will have to score each video and provide brief feedback before progressing to the next. As per Yeates et al,[23] examiners will have 4 weeks after the OSCE to complete video scoring.

## Data collection

Student scores (live and video performances) from each site will be collated and labelled with unique identifiers indicating (1) student, (2) site, (3) circuit, (4) station, (5) examiner, (6) examiner-cohort and (7) video or live performance. These data will be used to address all psychometric research questions.

To address research questions 1 and 6, researchers will develop an initial programme theory (IPT)[36] to orientate and focus subsequent data collection and analysis. To develop the IPT, researchers will consider prior research on VESCA, published experiences of international multi-institutional OSCE collaborations, formal theories which concern institutional adoption of innovations and the views of a range of experience assessment professionals.

Data will be collected iteratively, interspersed by analysis,[37] through a mixture of observation, individual interviews[38] and (where feasible) focus groups, supplemented by available process data. This, along with score data, will be triangulated across modalities to support validity.

Interviews will sample individuals from all relevant stakeholder groups at each site, focused on individuals who have participated in the research OSCE. While sampling requirements will be data driven, indicative numbers of each group from each site are students (n=4), examiners (n=4), simulated patients(n=3) and OSCE administrators(n=1–2). All individuals participating in the OSCE will be invited to be interviewed. If offers of participation exceed sampling needs, then participants will be selected to maximise sample representativeness. Recruitment will be performed by email. Participation will be voluntary. Participants will receive study information and asked to record their consent through an online consent form. Interviews will be conducted by members of the research team (PI, or research assistants), and are expected to last 45–60 min. Interviews will be conducted in-person in a private place or via Microsoft Teams. Interviews will be audio recorded and professionally transcribed. Interviews will be guided by a topic guide which will draw from the IPT and evolving theory and will be illustrated by practice-based examples where needed. The interview approach will be adapted to glean, refine and then consolidate emerging theory.[39]

Two researchers will observe the 'on-the-day' conduct of the OSCE in each participating medical school, using Realist ethnographic observation methods.[40] As far as feasible this will include: preparation for the OSCE, including station layout, equipment set-up, timing and scoring methods; conduct of the OSCE, including student flow around the circuits and observation of students examiners and simulated patients behaviour and interactions during and between station performances;

students and examiners interaction with filming; and participants' responses to both the OSCE and VESCA in breaks or after the OSCE is complete. Researchers' observations will be recorded through field notes which may be supplemented by examples of items or materials from the OSCE, diagrams or photographs.

Process data will be collected by researchers from each school depending on availability and may include participant recruitment data, score data, website metrics from examiner training materials and metrics related to video scoring by examiners.

## Patient and public involvement

Patients and members of the public have been involved throughout the VESCA programme of research which has led to this study. This has included establishing the priority of the research, reviewing plain English summaries, contributing to the design of the research, reviewing progress contributing to elements of the analysis and interpreting findings and discussing future directions. Members of the public are expected to contribute to dissemination activities.

## ANALYSIS

Similar analysis methods will be used for both questions. Audio recordings of interviews and focus groups will be professionally transcribed. Observation field notes, where available, will be incorporated into the dataset as will summaries of score data, participation rates and engagement metrics from on-line video scoring by examiners and video access metrics from the online feedback portal for students. Analysis will use the stages described by Papoutsi et al.[41] This begins by reading or considering each piece of data line by line to judge its relevance to the IPT. Next, where needed, decisions will be made about the trustworthiness of relevant data. Next, researchers will allocate initial conceptual labels. Conceptual labels will be derived both deductively from the IPT and inductively based on researchers' interpretation of emergent issues. Researchers will then consider whether each labelled concept can be interpreted to represent a context (C), a mechanism (M) or an outcome (O) and will look for data which provides information on the relatedness of Cs, Ms, and Os, so that they may be developed into Context-Mechanism-Outcome-Configurations (CMOC). Drawing on relevant data, researchers will then interpret how each CMOC relates to the programme theory and iteratively revise the programme theory as more and more CMOCs are developed. Interpretation will use the analytic processes of juxtaposition, reconciliation, adjudication and consolidation to explore discrepancies and resolve differences. Interpretation will also use retro-duction, combining both induction based on emergence from the data and deduction from the IPT in order to unearth mechanistic relations within CMOCs and the Programme theory.[42 43] Analysis will proceed iteratively, interspersed

with new data collection until a coherent and plausible programme theory is reached.

## Psychometric analyses (used for RQs 2–5, 7,8)

Research questions 2–5 will be addressed using Many Facet Rasch Modelling (MFRM), conducted using Facets by Winsteps.[44] The dependent variable for analyses will be denoted 'total score' and will be calculated for each student on each station by combining the scores for each domain. Categorical independent variables will be available for each station score, describing the student (unique ID number); station (station number); examiner (examiner ID); examiner cohort (ex-cohort ID); and site (institution ID). These data will be analyses using a four facet Rasch model, with facets of: (1) student, (2) station, (3) examiner cohort and (4) site.

To ensure data are adequate for MFRM analysis, research will assess the dimensionality, ordinality and model-fit of data. Dimensionality will be assessed using principle components analysis of model residuals with random parallel analysis using R studio for R.[45] Ordinality of the scale will be determined by examining the Rasch-Andrich thresholds supplied in FACETs output data (FACETS V.3.82.3 Winsteps, Western Australia). Fit parameters supplied by FACETs will be examined to determine data to model fit, using the criteria advocated by Linacre.[46] If data are inadequate for MFRM analysis, then the analysis plan will be adapted to use an appropriate alternative method such as linear mixed modelling.

To explore the potential that differences in institutional implementation of the OSCE might confound the measurement of examiner-cohort effects between institutions, we will additionally compare examiner cohort effects on the subset of score data arising from examiners' video scoring. This will offer a controlled comparison (as all examiner cohorts will score the same video performances). Analysis will use generalised linear modelling, including only data from examiners scoring of videos. The dependent variable will be total score, with factors of: station, examiner cohort, and school will be included in the model. Results from this analysis will be presented alongside the main analysis, to enable the likelihood of bias in the MFRM to be judged as part of overall evaluation of the complex intervention.

To address RQ2, observed (Raw score) average scores and 'Fair-Average' scores[47] for examiner cohorts will be compared, and the difference between observed (Raw score) average and Fair average will be calculated for each examiner cohort and compared. Observed differences will be transformed into multiples of the SE to calculate statistical significance.

To address RQ3 observed (Raw score) average scores and 'Fair-Average' scores[47] for each site (institution) will be compared and the difference between their observed (Raw score) average and Fair average will be calculated for each site and compared

To address RQ4, the difference between examiner cohorts within each institution (ie, site) will be calculated

and compared with the differences between the values for different institutions

To address RQ5a, the difference between the raw observed average score and the fair average score will be calculated for each participating student. These will be converted to mean absolute differences (MAD) to remove the direction of score adjustment. Descriptive statistics will be calculated for both the raw score adjustments and MAD adjustments. Similar to prior research,[22 24] the effect size of each MAD score adjustment will calculated using Cohen's d,[48] using the SD of students' average observed scores as the denominator. The mean Cohen's d and the proportion of students' whose adjustment exceeds d=0.5 will be reported.

To address RQ5b&c, category boundaries will be developed using the borderline regression method[49] for each station and pooled to give an average cut score for the test. Two separate values will be interpolated from the x-axis: one to represent a fail/pass boundary and one to represent a pass/excellent boundary. Each students' categorisation for the OSCE relative to these boundaries will be determined based on their observed raw average score and their fair average score and the proportion changing categories (number increasing a grade; number reducing a grade) will be calculated for both thresholds. Students rank position in the OSCE (regardless of institutional rankings) will be calculated based on observed raw average scores and fair average scores and the difference between each student's rank position from each score calculated. This will be expressed as both raw change in rank (positive or negative sign) and MAD change in rank which will be summarised through descriptive statistics.

Research questions 7 and 8 represent exploratory forms of analysis. These analyses will use the scores in individual scores domains within each station as dependent variables. Domains will be grouped based on content into dimensions which represent communication skills, knowledge and reasoning, investigation and management and procedural skills. Exploratory factor analysis will be used to determine the level of support for these dimensions, and Cronbach's alpha will be used to estimate the reliability of scores within each dimension. Student-level dimension scores will be examined to produce descriptive statistics describing dimension level scores and to determine the proportion of students who show greater than 0.5 SD score difference between difference dimensions. Further exploratory analyses will determine whether categorical differences exist for some students across domains (ie, greater frequency of borderline categories in one domain).

## Anticipated outcomes

Realist evaluations will produce mature programme theories which describe how different contexts elicit different mechanisms to produce varied outcomes for different stakeholders when (1) an integrated authentic task OSCE is shared between medical schools and (2) VESCA is implement across multiple medical schools. This will be

used to produce guidance on successful implementation of both interventions. Realist Evaluations will be reported using the standards of the RAMESES II (Realist And Meta-narrative Evidence Syntheses: Evolving Standards) reporting standards.[38]

Psychometric analyses for RQs 2–5 will describe the extent of overall score variability which arose between examiner cohorts and institutions in the standard of examiners' judgements, and the impact of adjustment for these on students' scores, categorisation and rank.

Psychometric analyses for RQs 7 and 8 will describe the dimensionality of domain-score data and varied patterns of strength and weakness in students' performances, with comparison in patterns across schools.

### Ethics and dissemination

This study will recruit volunteer students, examiners and SPs. Recruitment will invite the entirety of relevant students and examiner populations, subject to any local exclusions (eg, adequate academic progress). Simulated patients will participate as per their usual professional working arrangements. Participants will retain the right to withdraw up until their data are anonymised after which point withdrawal will not be possible. Researchers will collect personal data to manage recruitment and to link scores from the OSCE, online usage and engagement data for video access or scoring and interview and focus group data. These data will be stored securely and treated as confidential. Access will be limited to those members of the research team who require access for the analyses specified within the research. Participants will be asked to indicate whether they permit their data to be used in future research or to be contacted about future research. There are few anticipated risks to participants: if videos, score or interview data pertaining to them were disseminated inadvertently then that could cause embarrassment or distress. This risk is mitigated through the confidentiality and data security measures which will be employed. Students may benefit from taking part in the research through the experience of novel OSCE assessment tasks or availability of video feedback. Examiners may benefit from practice at examining. Ethical approval for the study has been granted by Keele University Research Ethics committee (Ref: MH-210209)

Study reporting will describe the blue printing and station development process; scoring format; an overview of station content and test reliability.

Findings of the research will be disseminated through academic publications, conference presentations and workshops and through engagement meetings with educational institutions who may adopt or implement VESCA or video-based feedback.

### Outputs

Good practice guidelines for the use of VESCA to enhance OSCE examiner standardisation in distributed exams and for sharing integrated task OSCEs across institutions. Intended audiences: institutions, assessment leads,

examiners. Engagement work through the Association for the Study of Medical Education Psychometrics Specialist Interest Group to promote this to policy makers.

Explanatory video, which will describe the purpose, use and benefits of VESCA for a lay audience. Intended will be audience, students, examiners, members of the public.

### Publications

The research is expected to produce academic publications describing the following findings:

1. Paper 1: primary psychometric analyses, comparing the influence of examiner cohorts and institutions on students' scores, categorisation and rank.
2. Paper 2: secondary psychometric analyses, determining the extent of additional diagnostic information available in domain score data.
3. Paper 3: realist evaluation, a programme theory of the implications of using integrated-task OSCE stations to increase authenticity in OSCE and using VESCA within a shared OSCE.

### Anticipated timeframe

Developing collaborations: complete by end May 2021.

Finalising protocol: June 2021.

Ethics application: July–September 2021.

OSCE station development: September–October 2021.

Scheduling and recruitment of OSCEs: October 2021–March 2022.

Site 1 OSCE: December–March 2022.

Sites 2–4 OSCE: January–July 2022.

Examiner video scoring: 4-week interval after each OSCE.

Interviews/focus groups/observations: December–August 2022.

Psychometric analyses: July–November 2022.

Realist analysis February–November 2022.

Dissemination: December 2022–February 2023.

**Author affiliations**

[1]School of Medicine, Keele University, Keele, UK

[2]School of Medicine, Dentistry and Biomedical Sciences, Queen's University Belfast, Belfast, UK

[3]School of Medicine, Cardiff University, Cardiff, UK

[4]School of Medicine, Medical Sciences and Nutrition, University of Aberdeen, Aberdeen, UK

[5]School of Medicine, University of Liverpool Faculty of Health and Life Sciences, Liverpool, UK

[6]Nuffield Department of Primary Care Health Sciences, University of Oxford Division of Public Health and Primary Health Care, Oxford, UK

**Contributors** The study design was developed by PY, RK, NC, GM, KC, VO, AC, RG, RV, C-WC, RKM and RF. PY wrote the original draft. GW provided expertise in Realist Evaluation methodology. PY, AM and NC are collecting data supported by KC, RV, C-WC, RG and RV. PY and AM will analyse the data. All authors critiqued and provided edits to the manuscript for intellectual content.

**Funding** The study is funded through a National Institute for Health Research (NIHR) Clinician Scientist award held by the Principal Investigator, Peter Yeates.

**Disclaimer** The study constitutes independent research and does not represent the views of the NIHR, the NHS or the Department of Health and Social Care.

**Competing interests** None declared.

**Patient and public involvement** Patients and/or the public were involved in the design, or conduct, or reporting, or dissemination plans of this research. Refer to the Methods section for further details.

**Patient consent for publication** Not applicable.

**Provenance and peer review** Not commissioned; externally peer reviewed.

**ORCID iDs**
Adriano Maluf http://orcid.org/0000-0001-8375-0533
Robert K McKinley http://orcid.org/0000-0002-3684-3435
Geoff Wong http://orcid.org/0000-0002-5384-4157

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
