## [Reviewer comments · BMJ Open]

ARTICLE DETAILS

TITLE (PROVISIONAL)	Enhancing Authenticity, Diagnosticity and Equivalence (AD-Equiv) in multi-centre OSCE exams in Health Professionals Education. Protocol for a Complex Intervention Study
AUTHORS	Yeates, Peter; Maluf, Adriano; Kinston, Ruth; Cope, Natalie; McCray, Gareth; Cullen, Kathy; O'Neill, Vikki; Cole, Aidan; Goodfellow, Rhian; Vallender, Rebecca; Chung, Ching-Wa; McKinley, Robert; Fuller, Richard; Wong, Geoff

VERSION 1 – REVIEW

REVIEWER	Schoenmakers, Birgitte KU Leuven, Public Health and Primary Care
REVIEW RETURNED	04-Jul-2022

GENERAL COMMENTS	Thank you for the opportunity to review this paper. The OSCE still is one of the most promising tools of assessment in medical education but has indeed some limitations, as you address most of them in this study protocol. But, what I miss overall is a renewing view on OSCE. The limitations of OSCE are well known and studied but until today not altered. Second, OSCE was initially developed as a formative, learning assessment and has now become a summative tool. To meet both considerations above, I believe that further studying the OSCE in its contemporary format does not meet the actual learning and assessment needs. Please, see also the other comments in the text. Hope this will help you to re-shape and re-focus the study protocol.
--

REVIEWER	Wyer, Mary Westmead Institute for Medical Research, Sydney Institute for Infectious Diseases
REVIEW RETURNED	12-Jul-2022

GENERAL COMMENTS	Thank you for the opportunity to review this manuscript entitled: Enhancing Authenticity, Diagnosticity and Equivalence (AD-Equiv) in multi-centre OSCE exams in Health Professionals Education. Protocol for a Complex Intervention Study." This is a well written paper that sets out its series of aims and research questions clearly. An appropriate and interesting intervention is described. The population, sample and recruitment strategy are well explained. Four sites have been scoped, but not yet confirmed. Data collection is supported with an easy-to-understand figure.
--

	It is pleasing to see that public involvement has been considered. Ethical considerations are attentive. Dates and timeline of the study have been included – as is a requirement of protocol papers for this journal. The CONSORT checklist for randomised trials was completed. The study is not randomised however section 1a of the checklist states that the title identifies randomisation. Elsewhere in the checklist randomisation questions are completed as “n/a”.
--	---

VERSION 1 – AUTHOR RESPONSE

Reviewers' comments/feedback	Authors' comments	Revised text
Reviewer 1		
Thank you for the opportunity to review this manuscript entitled: Enhancing Authenticity, Diagnosticity and Equivalence (AD-Equiv) in multi-centre OSCE exams in Health Professionals Education. Protocol for a Complex Intervention Study." This is a well written paper that sets out its series of aims and research questions clearly. An appropriate and interesting intervention is described. The population, sample and recruitment strategy are well explained. Four sites have been scoped, but not yet confirmed. Data collection is supported with an easy-to-understand figure. It is pleasing to see that public involvement has been considered. Ethical considerations are attentive. Dates and timeline of the study have been included – as is a requirement of protocol papers for this journal.	Thank you very much for these positive appraisals	No changes
The CONSORT checklist for randomised trials was completed. The study is not randomised however section 1a of the checklist states that the title identifies randomisation. Elsewhere in the checklist randomisation questions are completed as “n/a”	We apologise for this error. This has been changed to n/a to be consistent with the remaining entries.	

Reviewer 2:		
what I miss overall is a renewing view on OSCE. The limitations of OSCE are well known and studied but until today not altered. Second, OSCE was initially developed as a formative, learning assessment and has now become a summative tool. To meet both considerations above, I believe that further studying the OSCE in its contemporary format does not meet the actual learning and assessment needs.	Thank you for this comment. We agree that OSCEs have been extensively studied in the past. We suggest that this study does offer a renewing view on the OSCE, in that it seeks to respond to some of the substantial critiques of the OSCE (poor authenticity, examiner equivalence, lack of diagnosticity of students' strengths and weaknesses. Moreover, greater authenticity and diagnosticity offer a lot of potential to support students learning through more diagnostic feedback and the influence of more authentic tasks on students' preparation. We have added additional comment at the end of the introduction section regarding this.	Page 6: Collectively, it is anticipated that if these interventions are able to enhance the authenticity and equivalence of OSCEs whilst providing more diagnostic information on learners' performance, this will enhance OSCEs ability to support learning through their influence on students' preparation for OSCEs and their subsequent provision of more diagnostic feedback, whilst also ensuring greater confidence in the progression decisions which they inform. Consequently, understanding the interaction and use of these innovations is critical to determining their ability to benefit educational and healthcare practice.
U0031684 – see also https://www.ncbi.nlm.nih.gov/pmc/articles/PMC4224044/	Thank you for suggesting this reference. We have included it in the background section, at reference 15.	
This objective is too vague and mainly dependent on teaching and programs in the different institutions, rather than dependent on observer variability.	We are sorry that this objective was not sufficiently clear. We agree that sharing OSCE stations is likely to be influenced by the teaching and programs in different institutions. Indeed, that is precisely why we wish to study how the shared stations interact with the existing individual perceptions and assessment practices. We have added further specification to the objective to clarify this.	Page 6 “to share integrated-task OSCE stations across different institutions and understand the implications which arise from the interaction of these stations with existing individual perceptions and institutional assessment practices.”

U0031684 – This is not an objective but a condition to the study the above objectives	We disagree that this is not an objective. As we have described in the background, VESCA is a comparatively novel methodology which offers a method to compare and adjust for differences in examiners' scoring between locations. This objective 3 concerns using VESCA in the context of a shared integrated task OSCE to compare examiners judgements and to explore implications of cross-institutional use in this manner. This is distinct from Objective 2 which is to share and study integrated task OSCE stations. Consequently, whilst objective 2 is a necessary pre-condition for objective 3, objective 3 builds on and extends objective 2. We have added further signposting to highlight this progression.	Criticism 2: Examiner variability and challenges to equivalence.  • Objective 2: to share integrated-task OSCE stations across different institutions and understand the implications which arise from the interaction of these stations with existing individual perceptions and institutional assessment practices. Then, developing from that objective:  • Objective 3: to use the VESCA methodology within the context of a multi-centre integrated-task OSCE, to  a. compare and equate for differences between examiner-cohorts in different institutions and b. understand the implications which arise from using VESCA across institutions.
U0031684 – this is not a separate objective but a study of the underlying mechanism of objective 1	Again we disagree. Objective 1 is to develop station material which offers a more complex and realistic representation of practice and thereby provides a more authentic test of students' clinical skills. Traditionally, OSCEs result in a single score for each station which is aggregated to give a total competence score. This does little to highlight where students have focal areas of strength and weakness, which, in turn, offers little to guide further learning. This objective is distinct to the prior objectives in that it seeks to determine whether different domains of competence can be discerned from scoring domain data. If successful, these will provide information to focus support for learning and ensure that weaker areas can be addressed.	On page 5: Finally, recent inquiry has focused on ensuring that trainees are competent across all relevant domains of performance(27), with a view to both providing diagnostic information to support their learning and enabling focused areas of deficit to be addressed rather than simply demonstrating a sufficient total score, as is often the case in OSCEs (28). And: Criticism 3: Limited diagnosticity of OSCEs across different domains of performance.  • Objective 4: to determine whether different sub-domains of performance can be reliably distinguished from each other (rather than only providing an overall

	To clarify this, we have added text to the background, where this issue is explained, and slightly reworded the objective for clarity.	competence score) within a shared integrated-task OSCE.
U0031684 – How will you be able to rule out micro-bias such as input of simulated patients, circumstances/context, etc... with this small sample size	This point is important. We cannot rule out microbias due to the input of simulated patients, circumstances/context, and indeed our intention is to study whether and how these differences may arise between institutions due to differences in their culture and practice. We do not, however, believe that this is related to the sample size, as issues of institutional difference would be expected to produce a systematic rather than a random influence, which would therefore not be lessened by a larger sample. Instead we think it is important to study this potential to understand how it could occur and what influences it may produce. We have reflected on the potential for these differences to influence our quantitative comparison of examiners' scoring tendencies. Estimates of Examiner-Cohort effects have the potential to be confounded by differences in implementation between schools. All examiner groups, will however, score the same pool of videos of students' performances as part of this process. As these will be the same video performances, they offer controlled comparisons. As a result, we have proposed an additional analysis in which we will also compare examiner effects in this subset of data (video-based scores) and compare it to the results of the main analysis. We will present both these analyses to enable readers to judge the likelihood of bias within the overall evaluation of the complex intervention.	Page 2: (strengths and limitations section  • Whilst it is part of the object of study to explore how institutional differences in implementation might alter OSCE conditions, any such effects could potentially bias estimates of examiner-cohort effects in the main analysis. This is a limitation. The study's use of video-based comparison of examiners' scoring will enable controlled comparison of a subset of these responses, which will also be presented to enable the likelihood of such bias to be judged. Page 14: To explore the potential that differences in institutional implementation of the OSCE might confound the measurement of examiner-cohort effects between institutions, we will additionally compare examiner cohort effects on the subset of score data arising from examiners' video scoring. This will offer a controlled comparison (as all examiner cohorts will score the same video performances). Analysis will use generalised linear modelling (GLiM), including only data from examiners scoring of videos. The dependent variable will be total score, with factors of: station, examiner-cohort, and school will be included in the model. Results from this analysis will be presented alongside the main analysis, to enable the likelihood of bias in the MFRM to be judged as part of overall

	Consideration of the potential for bias has been added to the limitations section, and this additional analysis has been added to the analysis section.	evaluation of the complex intervention.
U0031684 – This section needs more attention: blueprint of OSCE? Construction of tasks? Modelling of score forms and scenarios	These sections have been expanded with greater detail of the construction of tasks, blueprinting and scoring format.	Page 9-10: For example, a station may describe a specific clinical scenario from the work of a new doctor and instruct candidates to perform a relevant clinical assessment. Candidates might then be expected to gather a clinical history, perform relevant focused physical examination, interpret provided investigation results, consult available guidelines and then describe their diagnosis and management to the patient. Tasks will be blueprinted against the UK General Medical Council's Clinical Skills Performance Assessment framework(34), to sample this framework's 3 domains: areas of clinical practice; clinical and professional capabilities; and areas of professional knowledge. Page 10: Examiners at all sites will score students' performances on the GeCoS rating system(35). This scoring system selects 5 appropriate performance domains for each station from a list of 20 when the station is designed (for example: history content, physical examination, clinical reasoning, building and maintaining the relationship, management content). Each domain is scored 1-4 (1=must improve; 2=borderline; 3=proficient; 4=very good). These scores are combined with a further 7-point global rating (1=incompetent; 7=excellent) to give a total score out of 27 for each station.

U0031684 – examination time will be 6*13.5 minutes which is below the threshold of reliability of OSCE (90 minutes)	Thank you for this comment. Students will be directly observed on each task for 13.5 minutes, but will spend further time preparing for the stations. As a result, the total testing time will be 90-105 minutes depending on each school's arrangements. This has been clarified in the text.	The OSCE will comprise six tasks (stations). In each station, students will be directly observed for 13.5 minutes, with a further variable amount of preparation and rotation time of between 1.5 – 4 minutes per station, depending on each school's usual practice. Consequently, total testing time will range between 90-105 mins depending on different schools' practices.
U0031684 - Videos for assessment will be distributed across institution	Yes, they will be. This has been clarified.	Page 11: All examiners will be asked to score the three selected videos selected for the station they examine. All examiners who examine a given station will score the same videos. To facilitate this, videos will be securely shared across institutions, using the secure on-line video scoring approach developed by Yeates et al(24). This will include the following elements: ...
U0031684 - how will you evaluate the construction of the OSCE? Content, blueprint, scenarios, score forms? Reliability? Validity?	Whilst evaluating these items is not central to our research questions, we agree that they are important quality markers which should be reported. We have clarified that we will report these factors.	Page 21: Study reporting will describe the blueprinting and station development process; scoring format; an overview of station content and test reliability.